# Microstructure and Texture Evolution with Relation to Mechanical Properties of Compared Symmetrically and Asymmetrically Cold Rolled Aluminum Alloy

**Jakob Kraner** [1,2,*], **Peter Fajfar** [2], **Heinz Palkowski** [3] , **Goran Kugler** [2], **Matjaž Godec** [1] and **Irena Paulin** [1]

1   Institute of Metals and Technology, Lepi pot 11, SI-1000 Ljubljana, Slovenia; matjaz.godec@imt.si (M.G.); irena.paulin@imt.si (I.P.)
2   Department of Materials and Metallurgy, Faculty of Natural Sciences and Engineering, University of Ljubljana, Aškerčeva cesta 12, SI-1000 Ljubljana, Slovenia; peter.fajfar@ntf.uni-lj.si (P.F.); goran.kugler@ntf.uni-lj.si (G.K.)
3   Department of Metal Forming and Processing, Institute of Metallurgy, Faculty of Natural and Materials Science, Clausthal University of Technology, Robert-Koch-Straße 42, DE-38678 Clausthal-Zellerfeld, Germany; heinz.palkowski@tu-clausthal.de
*   Correspondence: jakob.kraner@imt.si; Tel.: +386-1-4701-990

**Abstract:** The impact of asymmetric cold rolling was quantitatively assessed for an industrial aluminum alloy AA 5454. The asymmetric rolling resulted in lower rolling forces and higher strains compared to conventional symmetric rolling. In order to demonstrate the positive effect on the mechanical properties with asymmetric rolling, tensile tests, plastic-strain-ratio tests and hardness measurements were conducted. The improvements to the microstructure and the texture were observed with a light and scanning electron microscope; the latter making use of electron-backscatter diffraction. The result of the asymmetric rolling was a much lower planar anisotropy and a more homogeneous metal sheet with finer grains after annealing to the soft condition. The increased isotropy of the deformed and annealed aluminum sheet is a product of the texture heterogeneity and reduced volume fractions of separate texture components.

**Keywords:** asymmetric rolling; aluminum alloy; planar anisotropy; mechanical properties; microstructures

## 1. Introduction

Rolling is one of the most common metal-forming processes, frequently used for steels and aluminum alloys [1–5]. With their lower density and higher corrosion resistance, aluminum alloys will be used for increasing numbers of components in the automotive industry [6–8]. Of particular interest is the AA 5xxx series, because its mechanical properties have the potential to be improved by asymmetric rolling, endowing it with mechanical properties that are closer to those of steels [9–11].

Asymmetry in the rolling process can be introduced in different ways. Kinetics, geometry and friction are the major parameters that can be influenced to create the asymmetry. This can involve rolling with different roller diameters, rolling with the uneven use of lubricants, rolling with single-drive roller and rolling with different rotation speeds of the rollers. The latter two are more appropriate for rolling mills in the industry [12–17]. Rolling with different rotation speeds, where the so-called "grabbing problems" are less significant than with a single-drive roller, have an impact on changes to the workpiece's thickness as a consequence of the uneven distribution of the longitudinal velocity. Higher contact and shear stresses as well as friction differences on the contact surfaces were introduced while rolling with a speed difference between the upper and lower rollers of 5 m·s$^{-1}$. A larger difference

between the velocity of the upper and lower work roller contributes to a greater reduction of the rolling force [18–22]. At the same time, the speed difference results in the creation of a larger shear deformation area, which appears as a consequence of the changed position of the neutral points in the deformation zone [23,24]. On the other hand, a negative consequence of asymmetric rolling is the bending of the workpiece, more often known as the "ski effect" [25–27].

The asymmetric rolling of aluminum was in most cases investigated for thin sheets with small dimensions [28,29]. This means it would be interesting to study the impact of asymmetry in the rolling process on thicker sheets. In addition to achieving higher strains and creating a more homogeneous microstructure with smaller crystal grains, investigations of asymmetric rolling need to focus on the texture components. It would be particularly useful to know which of the rolling, shear and recrystallization texture components are the reason for the more effective heat treatments and the lower anisotropy [30–35]. For thinner sheets, the Erichsen test is normally used to assess the formability properties. In the case of thicker sheets, the plastic-strain-ratio test can be performed as an alternative. The calculated values of the so-called Lankford factor were in some cases, in previous investigations, much smaller for asymmetrically rolled samples than for symmetrically rolled samples, which means the formability is improved [36–39].

In our investigation the influences of asymmetric rolling in comparison to symmetric rolling were studied with respect to the technological, mechanical and metallographic perspectives. The interactions between the measured rolling forces, the achieved strains, the tensile and yield strength, the hardness, the indicators of planar anisotropy, the microstructures with an average size of grains and crystallographic textures with volume fractions of different rolling, shear and recrystallization texture components were investigated, with the aim to determine appropriate asymmetric rolling conditions for industrial applications.

## 2. Materials and Methods

### 2.1. Material and Geometry

The experimental materials were industrially produced via vertical direct chill casting with a subsequent homogenization of the slab, which was further hot rolled and coiled. A sheet with a thickness of 6.7 mm was cut into plates with an entry size of approximately 510 mm × 230 mm. The chemical composition of the AA 5454 aluminum alloy plates (Table 1) was controlled in the laboratory with X-ray fluorescence (XRF) and with inductively coupled plasma, optical emission spectrometry (ICP-OES).

**Table 1.** Chemical composition of AA 5454 aluminum alloy (wt.%).

| Mg | Mn | Fe | Si | Cr | Al |
|------|------|------|------|------|------|
| 2.43 | 0.61 | 0.25 | 0.18 | 0.09 | Bal. |

### 2.2. Rolling Parameters

The plates were rolled using two different roll gaps, i.e., 4.0 mm and 3.1 mm. With the roll gap of 4.0 mm the imposed strains were around 33%, and with the roll gap of 3.1 mm, around 44%. The rolling asymmetry results from the different rotation speeds of the two identical rollers of diameter 295 mm. In the laboratory experiments the rotation speed of the upper roller was kept constant at 10 rpm, while the rotation speed of the lower roller was set at 10 rpm, 15 rpm and 20 rpm. The corresponding asymmetry factors ($\omega_{lower}/\omega_{upper}$), as a consequence of the speed difference between the upper and lower rollers, were equal to 1.0, 1.5 and 2.0. The asymmetry factor 1.0 represents symmetric rolling, while the 1.5 and 2.0 factors of asymmetry represent two different types of asymmetric rolling. To recreate industrially relevant conditions a special lubricant, Somentor$^{TM}$ 32, was used. Minimally and for each rolling type same quantity of lubricant was added on both rollers, with intention to avoid the adhesion,

which is frequent for rolled aluminum. With lubrication at symmetric and asymmetric rolling the friction was reduced and consequently the factors of asymmetry were maintained. When lubrication applied, the factors of asymmetry are expected to be lower than the design value. On the other hand, the cold dry rolling is very rarely used and without lubrication the right response of material will not be reached and comparable between symmetric and asymmetric rolling with industry-similar conditions.

*2.3. Mechanical Tests*

The mechanical tests and the metallographic analyses were carried out on deformed and heat-treated samples. The heat treatment involved 400 °C for 1 h, which is common in the industry for this alloy. The mechanical properties from the tensile test, in accordance with the ASTM E8 M standard, and the plastic-strain-ratio test, in accordance with the ASTM E517 standard, were evaluated along the rolling direction (0°), the transverse direction (90°) and the diagonal direction (45°). The Brinell hardness was measured on the surfaces on the top and the bottom of the plates and in the centre position of the cross-section.

*2.4. Microscopy*

Prior to the light microscopy (LM) and the electron-backscatter diffraction (EBSD), the samples were mechanically grinded and polished. The last preparation step was polishing for 10 min with OPS and ion etching 50 min. For the metallographic analyses and the determination of the average size of the grains, in the top, bottom and centre positions of the specimen's cross-section, a ZEISS AXIO Imager M2m microscope (Carl Zeiss AG, Oberkochen, Germany) was used. A JEOL JSM-6500F field-emission scanning electron microscope (JEOL, Tokio, Japan) equipped with an HKL Nordlys II EBSD camera using Channel5 version 5.0.9.0 software (Oxford Instruments HKL, Hobro, Denmark) was used to detect the texture components in the symmetrically and asymmetrically rolled samples. Twelve major texture components in the face-centered cubic (fcc) material were searched for. Four rolling texture components (C, S, B and D), three shear texture components (H, E and F) and five recrystallized texture components (G, R, P, Q and Cube) were investigated and their volume fractions determined. The crystallographic planes and directions, as well as the angles in Euler space, are listed in Table 2. The instrument was operated at 15 kV and a 1.3 nA current for the EBSD analysis, with a tilting angle of 70°. Individual diffraction patterns were obtained together with mapping of the areas of interest. The detection was set to 5–7 bands, with $4 \times 4$ binning. For each sample, a map 1235 μm $\times$ 1005 μm, in the cross-section centre and bottom surfaces, which was in contact with the faster roller, was measured with a maximum step size of 5 μm. The volume fraction of the texture components was determined with a 10° deviation.

**Table 2.** Major texture components in face-centered cubic (fcc) material [30].

| Designation | Miller Indices {hkl} <uvw> | Euler Angles ($\phi_1$, $\Phi$, $\phi_2$) |
|:---:|:---:|:---:|
| B | {011} <211> | (35°, 45°, 0°) |
| S | {123} <634> | (59°, 34°, 65°) |
| C | {112} <111> | (90°, 35°, 45°) |
| D | {4411} <11118> | (90°, 27°, 45°) |
| H | {001} <110> | (0°, 0°, 45°) |
| E | {111} <110> | (60°, 54.7°, 45°) |
| F | {111} <112> | (90°, 54.7°, 45°) |
| Cube | {001} <100> | (0°, 0°, 0°) |
| G | {110} <001> | (0°, 45°, 0°) |
| R | {124} <211> | (53°, 36°, 60°) |
| P | {011} <112> | (65°, 45°, 0°) |
| Q | {013} <231> | (58°, 18°, 0°) |

*2.5. Numerical Simulation*

The commercial finite element package ABAQUS CAE 2018 (Dassault Systémes, Vélizy-Vallacoublay, France) was used with a model modified from an explicit three-dimensional rolling model. For numerical simulations all initial dimensions of workpieces and rolling mill were the same as at experimental laboratory rolling. The density (2690 kg·m$^{-3}$), modulus of elasticity (70.5 × 10$^9$ Pa), Poisson's ratio (0.33) and stress-strain values of initial (hot rolled) material, with the yield stress 151.0 MPa, were set as mechanical properties of specific aluminum alloy. The set time period was 2.5 s and the used friction coefficient were 0.1, 0.15 or 0.2, what is in accordance with the velocity of rollers. Rotation speed of rollers (0.1667, 0.2500 and 0.3333 radians·time$^{-1}$) and roll gap set (4.0 mm and 3.1 mm) were the same as at laboratory rolling, that way the factors of asymmetry and strain were approximately equal.

## 3. Results and Discussion

The results are presented as a comparison between symmetric and asymmetric rolling. The technological, mechanical and metallographic findings were examined with respect to the strain and the samples' condition, i.e., deformed or annealed.

*3.1. Rolling Force and Ski Effect*

An undesirable ski effect only appeared with the asymmetric rolling. Stated is clearly shown with the comparison of numerical simulations for symmetric (Figure 1a) and asymmetric (Figure 1b) rolling. The same phenomena were observed also at laboratory rolling. Comparing simulations and experiments the same functions, effects and phenomena were observed. Asymmetric rolling provided thinner rolled plates at the same roll gap set and that made the rolling process faster than symmetric rolling. With a higher factor of asymmetry exit thicknesses of plates were closer to the set roll gap, what was noted at simulations and laboratory rolling as well.

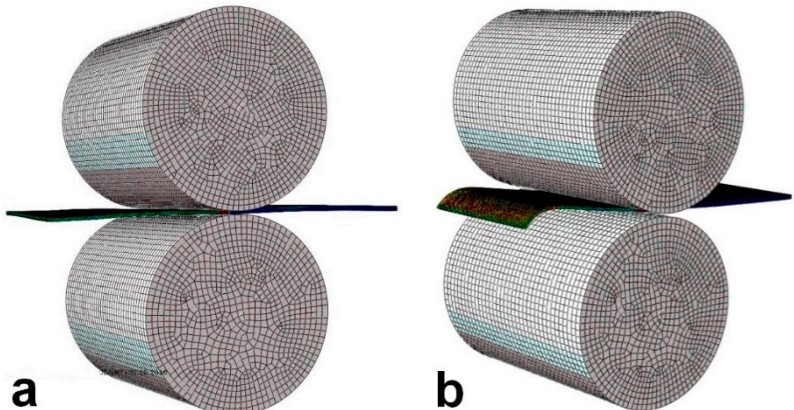

**Figure 1.** Numerical simulation of symmetric rolling (**a**) and asymmetric rolling (**b**).

Figure 2 shows the influence of the rolling-speed asymmetry factor on the strain of the plates and the rolling force for two set roll gaps of 4.0 mm and 3.1 mm. It is clear that with an increasing asymmetry factor the rolling forces were decreasing with the increasing strain. Explanation for that occurrence is in a much lower mean contact pressure at asymmetric rolling than in the case of symmetric rolling. That phenomena functioned by the denature shape of the deformation zone, which creates favorable circumstances for evolving longitudinal tensile stresses, whose action above the workpiece during the rolling match the previous and subsequent applied stress. Stated extensively reduces the mean pressure exerted on the contact surface of the rolls and as significance also rolling forces [22]. In the case of the symmetric rolling for a set roll gap of 4.0 mm, the rolling force was 1128 ± 3 kN at a 30.6 ± 0.6% strain. Compared to asymmetric rolling for the same set roll gap, the strain was

higher and the rolling forces were lower. The obtained rolling force for a 1.5 factor of asymmetry was 1099 ± 2 kN, and for a factor of asymmetry 2.0 it was 1083 ± 7 kN. The differences between the strains were not significant. The achieved strain for the factor of asymmetry 1.5 was 31.4 ± 0.5% and for the factor of asymmetry 2.0 it was 31.5 ± 0.2%. Somewhat smaller differences in the strains for the asymmetric rolling were obtained for the set roll gap of 3.1 mm. In both cases the strain was up to 44.9 ± 0.5%. In spite of the similar strain, a rolling force of 1341 ± 2 kN for the lower factor of asymmetry and 1309 ± 3 kN for the higher factor of asymmetry were measured. Compared to the symmetric rolling, a rolling force of 1377 ± 2 kN and a strain of 42.7 ± 0.1% were obtained. The increase in the strain and the decrease of the rolling force with asymmetric rolling was the case for both set roll gaps. For the higher set roll gap, the difference in the achieved strain between the symmetric and the asymmetric rolling process was around 2.5%. At a lower set roll gap the difference was 5%. According to the presented strains, the decrease of the rolling force in the case of a factor of asymmetry equal to 1.5 was 2.5% and for a factor of asymmetry equal to 2.0 it was 5%. In general, the increases of the achieved strains and the decreases of the rolling forces were not significant, but in the case of more passes, with asymmetric rolling, the desired thickness would be achieved with a smaller number of passes, which has an important economic impact for the rolling mill's function and wear.

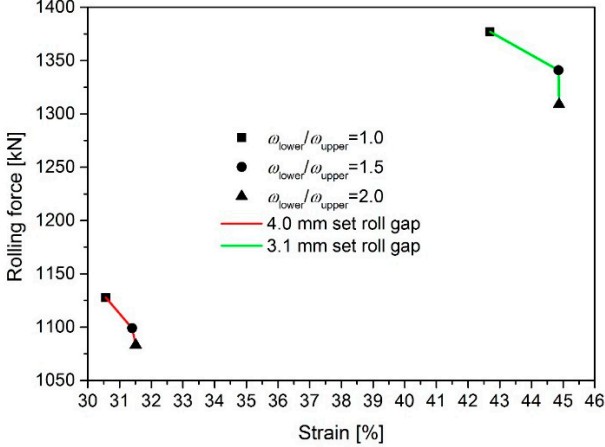

**Figure 2.** Measured rolling forces and calculated strains according to the two different roll gaps.

The ski effect only appeared on the asymmetrically rolled plates, and it was described using the length and the angle. The length of the ski effect was measured from the bended edge to the place where the plate became straight. The angle of the ski effect was defined as the highest angle of the plates' curvature. Ski effect on rolled plates is presented in Figure 3a. In Figure 3b the schematic presentation of the ski effect length and angle for separate rolling type is shown. Plates with exit dimensions from 684 mm to 785 mm had average values of the bended length from 57.0 to 63.6 mm (±3 mm). The differences between the measured angles of the bending area were small. At the same time, it should be noted that the factor of asymmetry had a higher impact on the formation of greater ski effect than the strain. Larger angles of 20.3 ± 0.5° and 21 ± 0°, as well as longer lengths of 60.3 ± 0.6 mm and 63.6 ± 1 mm, were measured on the plates rolled with a factor of asymmetry equal to 2.0. Shorter lengths of the ski effect equal to 57.0 ± 3 mm and 57.6 ± 0.6 mm were measured at a factor of asymmetry equal to 1.5. For this factor of asymmetry, the observed angles were between 18.3 ± 0.6° and 19 ± 0°. Asymmetric rolling with a faster lower roller results in a down-orientated ski effect. This can result in damage to the rolling table at the exit of the laboratory rolling mill or the transport rollers on industrial rolling mills, if in such cases asymmetric rolling is even possible. The ski effect represents a bent, useless area of rolled workpiece and must be cut off or made flat with mechanical intervention. The length of the ski effect, depending on selected asymmetric rolling type, will be significantly influenced by the vertical space between the deformation zone and the rolling table. The smaller the height between the workpiece on the exit and rolling table, the faster the bending process will finish and the workpiece

with a limited ski effect will slide forward on the rolling table. In the cases of asymmetric rolling from 7 ± 1% to 9 ± 1% of the ski effect according to the whole rolled plate were calculated. Comparing the symmetric and asymmetric plates, where, with the asymmetric rolling, between 2% and 5% more useless material was produced. On the other hand, it is necessary to emphasize that besides a larger percentage of a bent workpiece, the plates were longer because higher strains were reached with asymmetric rolling and the material will have better properties

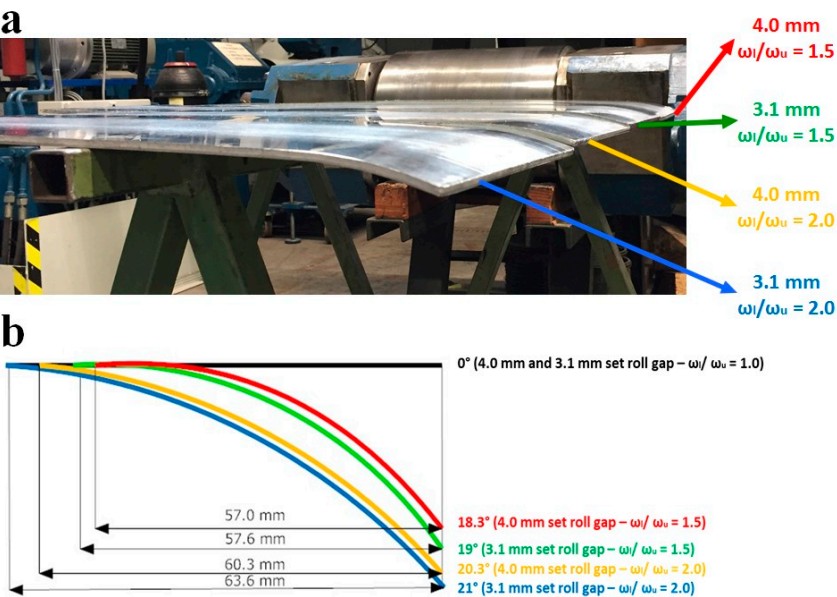

**Figure 3.** Ski effect on rolled plates (**a**) and schematic presentation of ski effect with measured lengths and angles (**b**).

### 3.2. Tensile and Yield Strength

With both set roll gaps the highest tensile-strength values were in the transverse direction. At the same time the tensile-strength values in the rolling direction with both set roll gaps were the lowest. For the 4.0 mm set roll gap the tensile-strength values were 279 ± 5 MPa for an asymmetry factor of 1.0 and 280 ± 5 MPa for asymmetry factors of 1.5 and 2.0. At a 3.1 mm set roll gap the average tensile strength values were higher. A factor of asymmetry of 1.0 brought us a tensile strength of 290 ± 6 MPa, with a factor of asymmetry of 1.5 the average value was 293 ± 6 MPa and for a factor of asymmetry 2.0 the tensile strength was 291 ± 7 MPa. Regarding the heat-treated samples for both set roll gaps, the highest values of the tensile strength were in the rolling direction. At the same time the lowest tensile-strength values after the heat treatment were observed in the diagonal direction. The average tensile-strength values after the heat treatment were similar for both set roll gaps and all three directions. Measured tensile strength was around 220 ± 3 MPa. In both cases of the roll-gap settings the highest values of the yield strength were in the rolling direction. The values for the factor of asymmetry equal to 2.0 were for both set roll gaps the highest and reached 235 ± 8 MPa with a set roll gap of 4.0 mm and 253 ± 1 MPa with a set roll gap of 3.1 mm. All the other yield-strength values for the deformed samples were for a higher set roll gap between 216 ± 6 MPa and 234 ± 5MPa and for a lower set roll gap between 244 ± 7 MPa and 251 ± 8 MPa. The highest yield-strength values were for a set roll gap of 3.1 mm with the heat-treated samples in the diagonal direction. All the values of the yield strength for heat-treated samples and all rolling types were lower and relatively even for both set roll gaps, with small deviations between 86 ± 0 MPa and 91 ± 1 MPa. The symmetrically and asymmetrically achieved strengths before and after heat treatment were very similar, with just smaller increases in both properties with asymmetric rolling. On the other hand, the tensile and yield strengths with asymmetric rolling stayed in the range of created strengths with symmetric rolling.

## 3.3. Anisotropy

For thicker sheets, the Erichsen test for deep-drawing properties could not be performed. For that reason the alternative plastic-strain-ratio test was conducted. The planar anisotropy indicator $\Delta r$ was calculated from the Lankford factor $r$. The values of $\Delta r$ could be positive or negative, depending on the relationship between the elongation and width increase of the tested sample. More important is that the $\Delta r$ values were close to 0 [38]. For all the rolling types and for both set heights of the roll gap, the $\Delta r$ values were negative (Table 3). The deformed sample's $\Delta r$ values for both set roll gaps were, in the case of asymmetric rolling, higher in the negative section than with symmetric rolling. Higher negative values of $\Delta r$ with the asymmetric rolling also mean a higher planar anisotropy, because the total isotropy of the material appears when $\Delta r$ is 0. Comparing the $\Delta r$ values of the deformed and heat-treated samples there was a strong contrast. The $\Delta r$ values for the deformed samples were decreasing with an increasing factor of asymmetry, and the values of $\Delta r$ for the heat-treated samples were increasing with an increasing factor of asymmetry. After the heat treatment $\Delta r$ was $-0.237 \pm 0$ for a factor of asymmetry equal to 1.0 at a 4.0 mm set roll gap. This was a more negative $\Delta r$ value than for the same sample in the just-deformed condition. The same phenomenon was observed with a factor of asymmetry equal to 1.5 and a larger set roll gap, where the $\Delta r$ after heat treatment was $-0.206 \pm 0.01$. For a factor of asymmetry equal to 2.0 $\Delta r$ was $-0.169 \pm 0.02$, which was less negative than the $\Delta r$ of the same, just-deformed sample. A more significant negative decrease in the values was visible with a smaller set roll gap. Of particular note is that for factors of asymmetry equal to 1.5 and 2.0 the $\Delta r$ values were $-0.052 \pm 0$ and $-0.050 \pm 0$. This is a very good indicator of the very low anisotropy for the asymmetric deformed samples after heat treatment, knowing that the $\Delta r$ after heat treatment with the same set roll gap and factor of asymmetry equal to 1.0 was $-0.112 \pm 0.02$. Regarding the higher $\Delta r$ values of the asymmetrically rolled samples in the deformed condition, after the heat treatment the asymmetrically rolled samples in all cases had a lower $\Delta r$ than the symmetrically rolled samples. This phenomenon could be attributed to a more appropriate texture for the successful heat treatment [29].

**Table 3.** Planar anisotropy indicator $\Delta r$ for deformed and heat-treated samples.

| Roll Gap Set | $\omega_{lower}/\omega_{upper}$ | $\Delta r$ Deformed Condition | $\Delta r$ Heat-Treated Condition |
|---|---|---|---|
| 4.0 mm | 1.0 | $-0.145 \pm 0.05$ | $-0.237 \pm 0$ |
| | 1.5 | $-0.197 \pm 0.05$ | $-0.206 \pm 0.01$ |
| | 2.0 | $-0.276 \pm 0.02$ | $-0.169 \pm 0.02$ |
| 3.1 mm | 1.0 | $-0.134 \pm 0.01$ | $-0.112 \pm 0.02$ |
| | 1.5 | $-0.157 \pm 0.02$ | $-0.052 \pm 0$ |
| | 2.0 | $-0.155 \pm 0.01$ | $-0.050 \pm 0$ |

## 3.4. Hardness

Hardness values for all rolling types and both conditions are presented in Table 4. The highest hardness values were obtained in the centre for all the rolling types. The values for the deformed condition with the larger set roll gap and factor of asymmetry equal to 1.0 were somehow higher than in the cases with a factor of asymmetry equal to 1.5 and 2.0. What is more important is that the difference between the surfaces hardness for the asymmetric types was smaller. The same phenomenon was observed with deformed samples with a set roll gap of 3.1 mm. The differences in the surface's hardness decreased in the case of the set roll gap with a higher factor of asymmetry. All the measured hardness values at the top, bottom and centre for heat-treated samples were more evenly spread than with the deformed samples. The differences in the surface hardness were smaller for the heat-treated samples than for the deformed samples. However, at the same time, it is clear that the hardness differences were smaller after the heat treatment of the asymmetrically rolled samples than for the symmetrically rolled samples.

**Table 4.** Hardness for all rolling types in deformed and heat-treated condition.

| Roll gap Set (mm; Factor of Asymmetry) | Measuring Position | Deformed Conditio (HB) | Heat-Treated Condition (HB) |
|---|---|---|---|
| 4.0 (1.0) | Top | 86 ± 0.5 | 57 ± 2 |
| | Centre | 90 ± 2 | 56 ± 3 |
| | Bottom | 83 ± 1 | 57 ± 2 |
| 4.0 (1.5) | Top | 83 ± 0.5 | 57 ± 1 |
| | Centre | 88 ± 1 | 57 ± 1 |
| | Bottom | 82 ± 3 | 56 ± 1 |
| 4.0 (2.0) | Top | 83 ± 1 | 57 ± 0.5 |
| | Centre | 88 ± 2 | 56 ± 0.5 |
| | Bottom | 80 ± 3 | 58 ± 1 |
| 3.1 (1.0) | Top | 84 ± 1 | 56 ± 2 |
| | Centre | 92 ± 0.5 | 58 ± 1 |
| | Bottom | 85 ± 2 | 57 ± 0.5 |
| 3.1 (1.5) | Top | 87 ± 2 | 58 ± 3 |
| | Centre | 92 ± 1 | 56 ± 1 |
| | Bottom | 86 ± 2 | 58 ± 3 |
| 3.1 (2.0) | Top | 86 ± 1 | 56 ± 1 |
| | Centre | 92 ± 1 | 56 ± 1 |
| | Bottom | 86 ± 0.5 | 56 ± 0.5 |

*3.5. Microstructure*

Comparing the microstructures of the deformed samples (Figure 4a–c) with the microstructures of the heat-treated samples (Figure 4d–f), some differences were observed. Since higher strains were achieved with the asymmetric rolling, the average size of the grains, as a consequence, was smaller. The centre band of the longitudinally deformed grains was for the asymmetrically rolled microstructure smaller and it was decreased with a higher factor of asymmetry. Moreover, it is important that the grains in the asymmetrically deformed samples are, in all three measured sections, more evenly distributed than for the symmetrically rolled samples. The results for the higher and lower set roll gaps are presented in Figure 5a,b. In the deformed samples the grain sizes are, for a set roll gap of 4.0 mm, between 26.4 ± 0.7 μm and 17.7 ± 0.5 μm, and for a set roll gap of 3.1 mm, between 23.3 ± 0.3 μm and 17.5 ± 0.6 μm. The grains in the centre position of the cross-section are, for all three different factors of asymmetry, larger than those on the top or bottom position. The difference between the grain size on the top and bottom positions is very small, but at the same time the smaller grains were obtained at the bottom position, which can be the impact of the higher velocity of the lower roller. This is valid for the deformed samples as well as for heat-treated samples. After the heat treatment, there is no major difference in the distribution of the microstructural components for all three sections. The grains in the microstructure are very even, especially with the asymmetrically rolled samples. Some impacts of the asymmetry remained after the treatment to the soft condition. For the heat-treated samples the grain sizes were between 33.9 ± 1 μm and 25.3 ± 0.7 μm for the set roll gap of 4.0 mm and from 25.0 ± 0.5 μm to 19.3 ± 0.6 μm for the set roll gap of 3.1 mm at the top, centre and bottom positions of the samples' cross sections. The presented average grain sizes for the heat-treated samples are shown in Figure 5c,d. A major advantage of the asymmetric rolling is the creation of more homogeneous material, in our case, confirmed with the average size of the grains and the hardness throughout the cross-section. More homogeneous material, i.e., the material with the more evenly distributed properties throughout the cross-section, is better for further forming processes, heat treatment and quality of the products [30,31].

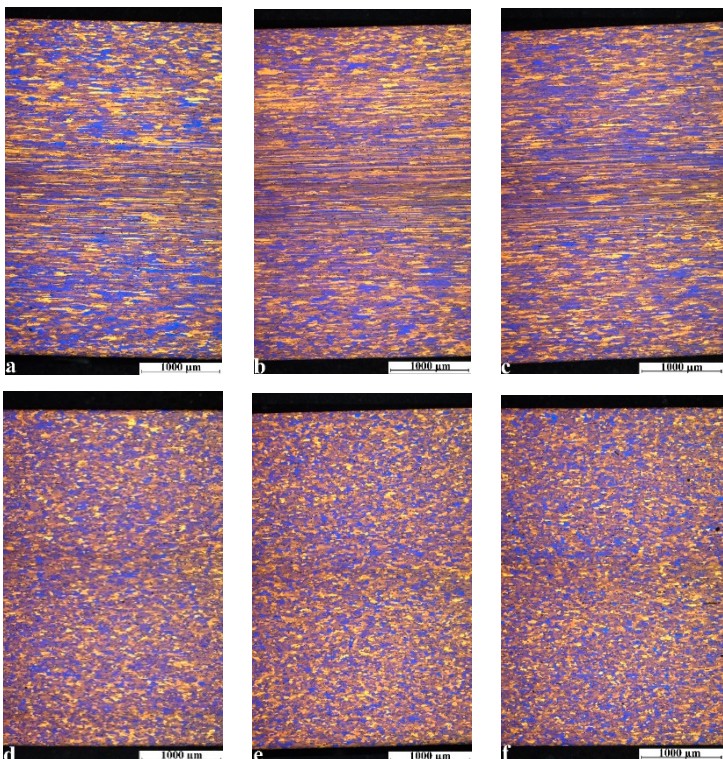

**Figure 4.** Microstructure in samples' cross-sections for symmetrically rolled 1.0 (**a**), asymmetrically rolled 1.5 (**b**), asymmetrically rolled 2.0 (**c**), symmetrically rolled 1.0 and heat treated (**d**), asymmetrically rolled 1. 5 and heat treated (**e**) and asymmetrically rolled 2.0 and heat treated (**f**).

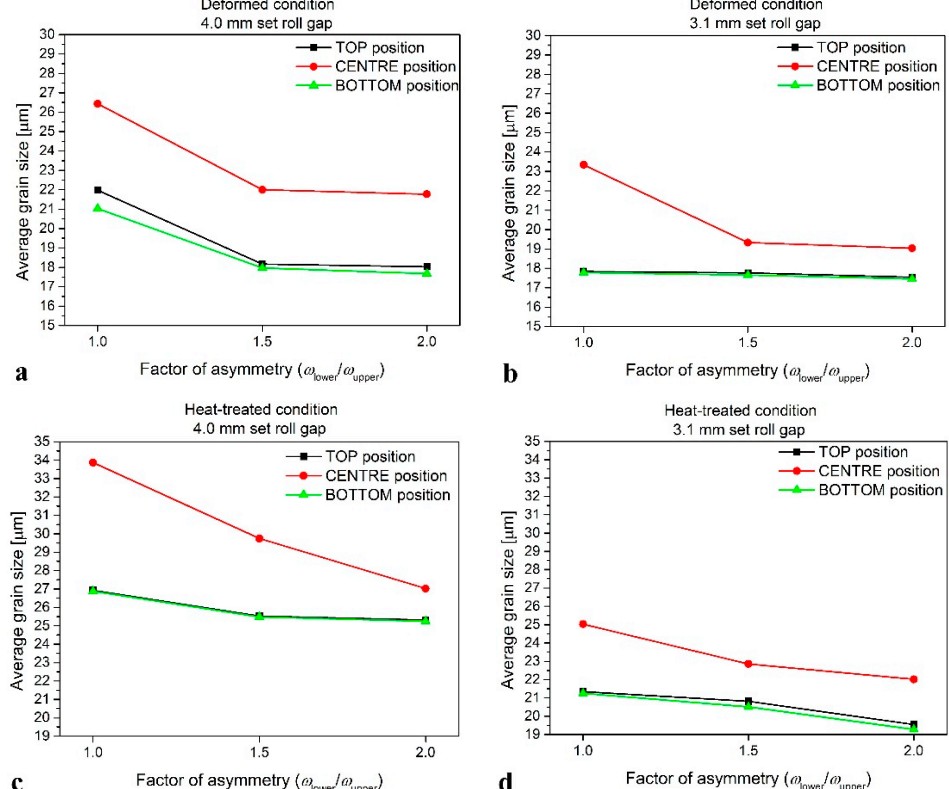

**Figure 5.** Average grain size for deformed condition with 4.0 mm set roll gap (**a**), deformed condition with 3.1 mm set roll gap (**b**), heat treated with 4.0 mm set roll gap (**c**) and heat treated with 3.1 mm set roll gap (**d**).

*3.6. Texture*

The texture components were detected with EBSD for the deformed and heat-treated samples. At the same position and for the same asymmetric rolling type, the deformed condition, with its strongly pronounced texture pattern and higher intensity of the pole figure, and heat-treated condition, with an even distribution of all the texture components and a very low intensity of the pole figure were observed. The EBSD mapping confirms the refinement of the crystal grains with asymmetric rolling, were with the higher factor of asymmetry the smaller grains were produced. The associated pole figures in Figure 6a–c present a lower intensity and that way also the higher heterogeneity of texture created with the asymmetric rolling. Observing the central position of the lower deformed and heat-treated sample, the texture component E at a factor of asymmetry equal to 1.0 and a texture component R at a factor of asymmetry equal to 2.0 stand out. All the other observed texture components for all three factors of asymmetry were from 0.0 to 2.5 vol.%. These results are presented in Figure 7a. The same figure shows that the volume fraction of all the rolling texture components in the case of asymmetric rolling is higher than for the symmetric rolling type. At the same time, a smaller difference in the volume fractions was observed between the separate texture components in the asymmetric rolling textures. The same phenomenon appears in Figure 7b, where the centre positions of the more deformed samples are presented. In the sample that was deformed with a factor of asymmetry equal to 1.0 and heat-treated higher volume fractions for all the rolling texture components were obtained in comparison to the asymmetric rolling types with factors of asymmetry equal to 1.5 and 2.0. Moreover, in the symmetrically rolled sample, the separate texture components were more obvious. This indicates that the volume fractions of all the texture components with asymmetric rolling were more even than for the symmetric rolling. In general, the volume fractions of all the observed texture components were, for the asymmetrically rolled and heat-treated samples with a set roll gap of 3.1 mm, between 0.1 and 1.8 vol.%. Closer to the faster roller, i.e., the bottom position of the sample, were with a set roll gap of 4.0 mm in a symmetrically rolled sample, there was the volume fraction of texture component E, with 6.3 vol.% and texture component G, with 3.9 vol.%. The highest volume fraction of the rolling texture components was, in Figure 7c, observed for a factor of asymmetry equal to 2.0. In Figure 7d, where the bottom section of the more symmetrically deformed and heat-treated sample was presented, the highest volume fraction of 3.1 vol.% was observed for the texture component R. The higher volume fractions of the rolling and shear texture components were obtained in this sample with a factor of asymmetry equal to 1.5. All the compared textures had a very even distribution of texture components, but at the same time, because the symmetrically rolled sample had more stood out recrystallization texture components, and the volume fractions of all the textures were, for the asymmetrically rolled samples with a factor of asymmetry equal to 1.5 and 2.0, significantly equivalent. For comparison of the observed texture components and the texture evolution the deformation history and the initial textures of the material are important [35]. The fibers as texture elements connected to different texture components must be exposed as a possible preference for successful heat treatment [31]. Nevertheless, a comparison between asymmetrically and symmetrically rolled textures is useful and can explain some changes in the mechanical properties. This is in addition to the detection of the C, S and B texture components, which could indicate the formation of β fibers as major texture element for highly deformed rolling materials with fcc.

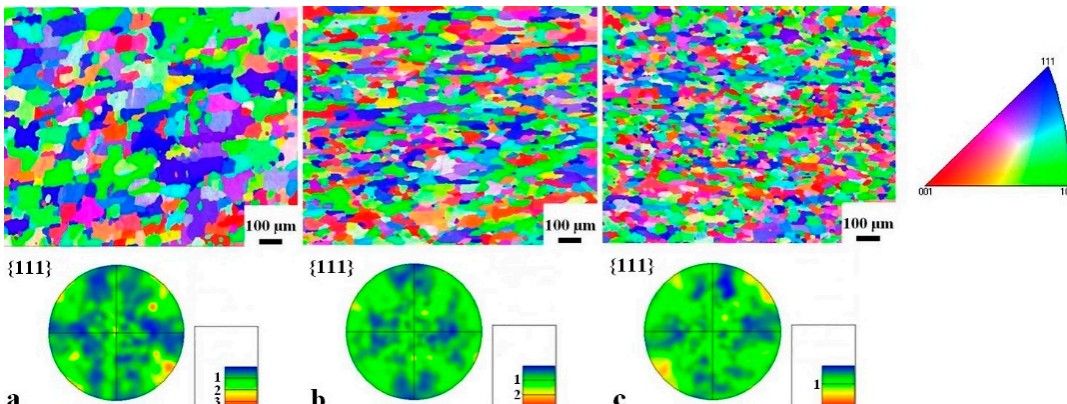

**Figure 6.** EBSD mapping (IPF Z) and pole figures for rolled and heat-treated samples in centre position for symmetrically rolled 1.0 (**a**), asymmetrically rolled 1.5 (**b**) and asymmetrically rolled 2.0 (**c**).

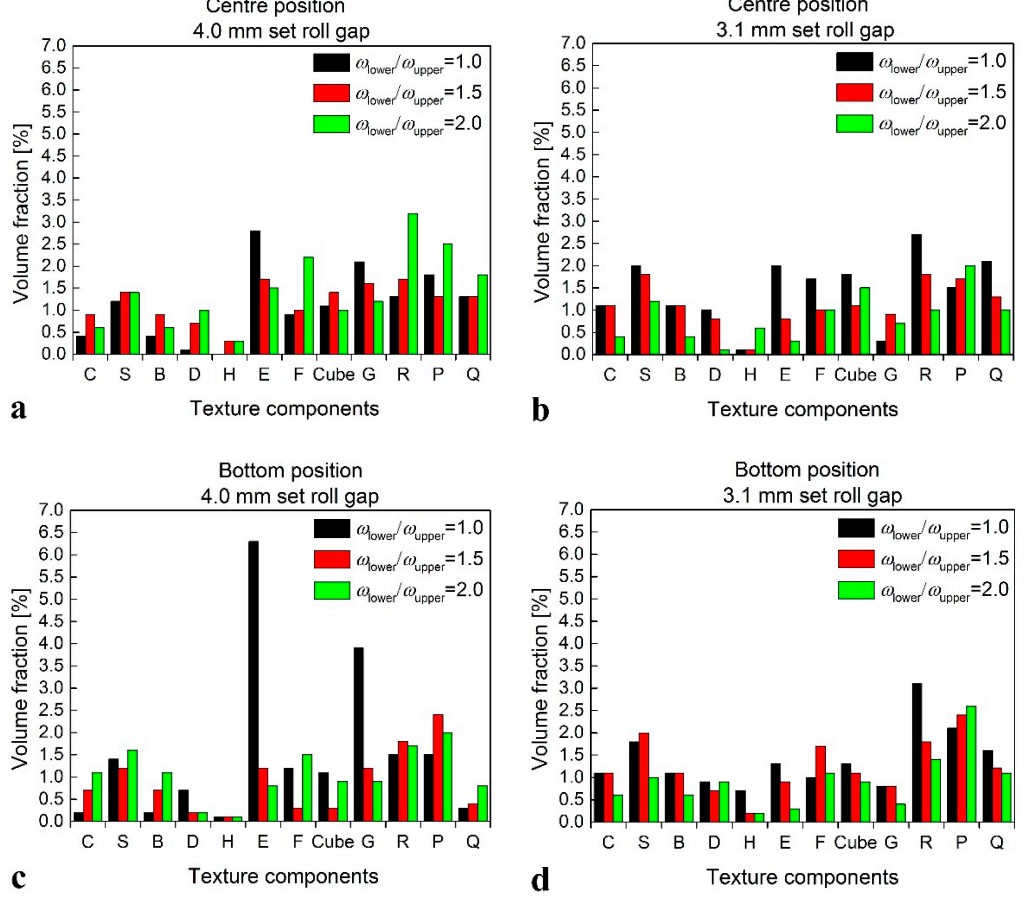

**Figure 7.** Volume fraction of texture components in heat-treated samples for 4.0 mm set roll gap and centre position (**a**), 3.1 mm set roll gap and centre position (**b**), 4.0 mm set roll gap and bottom position (**c**) and 3.1 mm set roll gap and centre position (**d**).

## 4. Conclusions

The asymmetric cold rolling of AA 5454 aluminum alloy plates was compared with symmetric rolling. Due to the uneven distribution of the upper and lower roller pressures and the workpiece velocity, the mechanisms in the deformation zone were more suitable for producing smaller crystal grains, which has an impact on improving most mechanical properties. The main advantage of asymmetric rolling is the presence of significant shear strain, leading to a gradient structure. At the

same time, the asymmetric rolling enables the creation of more shear texture components, in addition to the rolling and recrystallized texture components. The texture heterogeneity presented with more different texture components and less stands out volume fraction of a separate texture component will have lower planar anisotropy.

The asymmetric conditions were created by using different roller speeds. The general aim was to determine the impact of the asymmetry on the rolling process for a specific alloy. At the same time, metallographic analyses were made to explain some changes to the technological and mechanical properties. For asymmetric rolling, a lower rolling force was needed to create higher strains at the same set roll gap. Thickness reductions as well as strains increased and the rolling forces were decreased with a higher factor of asymmetry. The lengths and the angles of the ski effects were very similar, nevertheless, it was clear that the ski effect depends much more on the factor of asymmetry than the strain. The highest values of the tensile and yield strengths were not present in the deformed and heat-treated condition in the same direction. The elongations after the heat treatment were quite similar, which indicates an effective heat treatment.

The planar anisotropy in the deformed condition of the asymmetrically rolled samples was higher than with the symmetrically rolled samples. The elimination of the planar anisotropy after the heat treatment was larger in the asymmetrically rolled samples. The production of a more homogeneous material is visible in the samples with the more evenly distributed hardness throughout the cross-section, with smaller deviations in the case of the asymmetrically rolled samples than with the symmetrically rolled samples. The impact of the asymmetry is through the hardness of the material, also shown after the heat treatment. In accordance with the homogeneity of the hardness is the average size of the grains, where with a higher factor of asymmetry grains decreased. The grain size deviations throughout the cross-section were also lower with the asymmetrically rolled samples. Different crystallographic textures were detected at the central position, as well as at the bottom position of the same samples. The volume fractions of the detected texture components were, for the asymmetrically rolled samples, more evenly distributed than for the symmetrically rolled samples, where more separate texture components stand out.

**Author Contributions:** Conceptualization, J.K. and P.F.; methodology, J.K., P.F. and H.P.; software, J.K., M.G. and I.P.; validation, J.K., P.F., H.P., M.G. and I.P.; formal analysis, J.K., H.P., M.G. and I.P.; investigation, J.K., P.F. and H.P.; data curation, G.K.; writing—original draft preparation, J.K., P.F. and I.P.; writing—review and editing, H.P., M.G. and G.K.; supervision, P.F., H.P., M.G. and I.P.; funding acquisition, H.P., G.K. and M.G. All authors have read and agreed to the published version of the manuscript.

**Funding:** The authors acknowledge the financial support from the Slovenian Research Agency, research corefundings No. P2-0132 and No. P2-0344. This work was also funded by the program MARTINA, supported by Slovenian Ministry of education, science and sport and European Regional Development Fund.

**Conflicts of Interest:** The authors declare no conflict of interest.

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
