# Peer review of "Microstructure and Texture Evolution with Relation to Mechanical Properties of Compared Symmetrically and Asymmetrically Cold Rolled Aluminum Alloy"

_metals, doi:10.3390/met10020156_

Round 1

Reviewer 1 Report

In section 2.2, the effect of lubrication on the asymmetric rolling need to be  explained in more details. It is important to ensure asymmetry factors are maintained when lubrication is applied. The  numerical simulation setup need to be described in the experimental section. Figure 2 showed that the rolling forces were lower in the asymmetric rolling than those in the symmetric rolling. However, no explanation was provided. A comparison between the experiments and numerical simulation is needed for better understanding the effect of asymmetric rolling. It was unclear how the "ski effect" was measured in page 5. Hence, optical images of rolled plates with measured angles noted are necessary to demonstrate how severe the "ski" effect is. Line 161-163, This sentence need to be rewritten. Error bars are required for all measured values. Particularly for the tensile strength values, the difference between the symmetric and asymmetric rolling are trivial. Hence, error bars are essential to demonstrate whether these differences are meaningful. The description of hardness value is very vague in section 3.4. The hardness values at surfaces and center need to be listed for all conditions in a table. The texture measurements for the as-rolled samples (without heat-treatment) are required, as they are more valuable than those for the deformed and heat-treated samples.

Author Response

Dear editors and reviewers:

Thank you very much for your careful review and constructive suggestions with regard to our manuscript “MICROSTRUCTURE AND TEXTURE EVOLUTION WITH RELATION TO MECHANICAL PROPERTIES OF COMPARED SYMMETRICALLY AND ASYMMETRICALLY COLD ROLLED ALUMINIUM ALLOY”. We have studied comments carefully and tried our best to revise and improve the manuscript according to the referees’ good comments. The main corrections in the paper and the responds to the reviewer’s comments are as following:

LUBRICATION

The effect of lubrication was explained in section 2.2. There is also explained why dry rolling weren’t perform.

NUMERICAL SIMULATIONS SETUP

Research has much more attention and emphasis on experimental laboratory work as on numerical simulations. Nevertheless in manuscript (section 2.5.) was added that all workpiece and rolling mill dimensions as also most of needed mechanical properties of material were the same at modelling simulation as well as at laboratory rolling. 

LOWER ROLLING FORCES

For results in Figure 2 the explanation of rolling force reduction and correlation to the strain increase was added. 

COMPARISON BETWEEN NUMERICAL SIMULATIONS AND EXPERIMENTS

It was added that same effects and phenomena were observed as function processes at both researching types. Research has much more attention on experimental work then on numerical simulations, which were made just for examination of basic processes and effects.

SKI EFFECT

The definition of our length and angle measurements were added, as also the figure where the ski effect on rolled plates and scheme with values were presented.

LINE 161-163

Sentence was rewritten.

ERROR BARS

For each measured value the deviation (error bar) is written in the text. The same was done also in tables with results.

HARDNESS

Section 3.4 was improved and made more understandable. A table with listed hardness values at surfaces and centre was added.  

TEXTURES of AS ROLLED MATERIAL

To reach the best indexation with EBSD at deformed samples is much harder than at heat-treated (recrystallized) samples and with low indexation is difficult to claim, which texture components are in the texture certainly present. Almost never the products or semi-products are used in just deformed condition, so the textures in deformed condition can’t be more valuable part of investigation than textures in heat-treated condition. Furthermore, we observed the significant improvement of planar anisotropy at asymmetric rolling after heat treatment to the soft condition, so our attention was more in comparison of heat-treated samples textures at different positions in cross section. That is more valuable and has important meaning for correlation between properties and textures for industry alloys.         

Once again, we appreciate for Editors/Reviewers’ warm work earnestly, and hope that the corrections will meet with approval.

Yours sincerely,

Jakob Kraner

Institute of Metals and Technology

Reviewer 2 Report

The topic of work is relevant. In fact, there has recently been an increased interest in asymmetric rolling processes. The main feature of asymmetric rolling process is the rotation of the deformation zone caused by the actions of oppositely directed friction forces in the mixed zone. Asymmetric rolling provides a simultaneous pure and simple shear in the processed material and it induces higher strain in material at lower rolling force. Asymmetric rolling also makes crystal grains smaller and it leads to improvement of crystallographic textures. But these statements are well known and have already discussed in a lot of publications.

Some comments and questions:

1) The objective of this manuscript is not clearly defined.

2) The novelty of this manuscript must be clearly formulated.

3) Why only two asymmetry factors (1.5 and 2.0) and two thickness reductions were examined?

4) If initial thickness of the sheet was 6.7 mm and final thicknesses were 4.0 mm and 3.1 mm, then thickness reductions corresponded to 40.3 % and 53.7 % accordingly. How the strains around 33 % and 44 % were determined?

5) Shear strain is created by opposite friction forces, which should be very high during asymmetric rolling. Why lubricant was used in laboratory experiments? Lubricant can be used to recreate industrially relevant conditions of conventional symmetric rolling. But asymmetric rolling process is very sensitive to friction conditions and should be carry out without lubrication (on dry work rolls).

Author Response

Dear editors and reviewers:

Thank you very much for your careful review and constructive suggestions with regard to our manuscript “MICROSTRUCTURE AND TEXTURE EVOLUTION WITH RELATION TO MECHANICAL PROPERTIES OF COMPARED SYMMETRICALLY AND ASYMMETRICALLY COLD ROLLED ALUMINIUM ALLOY”. We have studied comments carefully and tried our best to revise and improve the manuscript according to the referees’ good comments. The main corrections in the paper and the responds to the reviewer’s comments are as following:

OBJECTIVE

The objective of manuscript is now defined in the introduction. Aluminium alloys are every year more and more represented in all areas of transport industry. For example, the AA 5454 is mostly used for train bodies. With asymmetric rolling, the formability and deep-drawing properties, which are worse in comparison to the steel, needs to be improved, so that aluminium alloys can be even more competitive for transport applications. Furthermore, the objective was directly connected with researching the significance for the usage in industry, with purpose that asymmetric rolling doesn’t just stay as promising laboratory method.   

NOVELTY

The novelty of manuscript is formulated in conclusions. In deformed condition the planar anisotropy is higher for asymmetrically rolled samples. Furthermore, for both different deformations the increase of planar anisotropy with higher factor of asymmetry was observed. The influence and improvement of planar anisotropy with asymmetric rolling appeared after heat treatment. Planar anisotropy is with higher factor of asymmetry lower, what was reached with both set roll gaps. From results which are as novelty presented in our manuscript is also clearly presented how imported is the choice of appropriate combination of strain (roll gap set) and factor of asymmetry. The significant reduction of planar anisotropy was reached with lower roll gap set and higher factor of asymmetry.

Why only two asymmetry factors (1.5 and 2.0) and two thickness reductions were examined?

Basic of that investigation was comparison between symmetric and asymmetric rolling. With that purpose we choose simple sequence of asymmetry factors (1.0, 1.5 and 2.0) to observe the influences of asymmetric rolling. To confirm trend, furthermore we execute the same roll types with lower set roll gap. That way, more comparison between different rolling parameters can be made. At this point we conclude that investigation has a clear sense and enough results to understand and present microstructure and texture evolution with relations to the mechanical properties of industrial aluminium alloy, when cold symmetric and asymmetric rolling are compared.     

STRAIN

The initial thickness of sheet was 6.7 mm, but 4.0 mm and 3.1 mm were just roll gap sets and not the final thickness of rolled plates. We didn’t roll with different types to define final thickness, but with the same roll gap set for symmetric and two asymmetric rolling types, what enable the real comparison of rolling force and strain (at the same time) between symmetric and asymmetric rolling. Because of the jump of rollers the final thickness was greater than the roll gap set and strains were smaller than your calculation (40.3 % and 53.7 %). The strains were around 33 % for 4.0 mm and 44 % for 3.1 mm roll gap set. Exact strain values for separate rolling type are presented in section 3.1 and shown in figure 2.      

LUBRICATION

In section 2.2 the lubrication reason and performance is described more in details. The explanations about the effect of lubrication and why dry rolling wasn’t perform are added.

Once again, we appreciate for Editors/Reviewers’ warm work earnestly, and we hope that the correction will meet with approval.

Yours sincerely,

Jakob Kraner

Institute of Metals and Technology

Round 2

Reviewer 1 Report

Page 2, Line87-89. With lubrication applied, the factors of asymmetry are expected to be lower than the design value. This need to be corrected in the text.

Author Response

Dear editors and reviewers: 

Thank you very much for your second careful review and constructive suggestions with regard to our manuscript “MICROSTRUCTURE AND TEXTURE EVOLUTION WITH RELATION TO MECHANICAL PROPERTIES OF COMPARED SYMMETRICALLY AND ASYMMETRICALLY COLD ROLLED ALUMINIUM ALLOY”. The minor corrections in the paper and the responds to the reviewer’s comments are as following: 

LINE 87-89

The wished sentence in text was added and that way the meaning was corrected.        

Once again, we appreciate for Editors/Reviewers’ warm work earnestly, and hope that the correction will meet with approval.

Yours sincerely 

Jakob Kraner

Institute of Metals and Technology

Reviewer 2 Report

In the process of responding to comments, the work has been improved and can be recommended for publication.
The main comment on the work is as follows:
The main advantage of asymmetric rolling is the presence of significant shear strain, leading to a gradient structure. In the proposed deformation modes, this advantage is practically not used.
Nevertheless, the work can be recommended for publication.

Author Response

Dear editors and reviewers: 

Thank you very much for your second careful review and constructive suggestions with regard to our manuscript “MICROSTRUCTURE AND TEXTURE EVOLUTION WITH RELATION TO MECHANICAL PROPERTIES OF COMPARED SYMMETRICALLY AND ASYMMETRICALLY COLD ROLLED ALUMINIUM ALLOY”. The minor corrections in the paper and the responds to the reviewer’s comments are as following: 

MAIN ADVANTAGE

The sentence about main advantage was added in conclusions.      

Once again, we appreciate for Editors/Reviewers’ warm work earnestly, and hope that the correction will meet with approval.

Yours sincerely 

Jakob Kraner

Institute of Metals and Technology